# Weakly-Supervised Audio-Visual Segmentation

**Shentong Mo**[1,2] **Bhiksha Raj**[1,2]
[1]CMU, [2]MBZUAI

## Abstract

Audio-visual segmentation is a challenging task that aims to predict pixel-level masks for sound sources in a video. Previous work applied a comprehensive manually designed architecture with countless pixel-wise accurate masks as supervision. However, these pixel-level masks are expensive and not available in all cases. In this work, we aim to simplify the supervision as the instance-level annotation, *i.e.*, weakly-supervised audio-visual segmentation. We present a novel Weakly-Supervised Audio-Visual Segmentation framework, namely WS-AVS, that can learn multi-scale audio-visual alignment with multi-scale multiple-instance contrastive learning for audio-visual segmentation. Extensive experiments on AVS-Bench demonstrate the effectiveness of our WS-AVS in the weakly-supervised audio-visual segmentation of single-source and multi-source scenarios.

## 1 Introduction

When we hear a dog barking, we are naturally aware of where the dog is in the room due to the strong correspondence between audio signals and visual objects in the world. This human perception intelligence attracts many researchers to explore audio-visual joint learning for visual sound localization and segmentation. In this work, we aim to segment sound source masks from both frames and audio in a video without relying on expensive annotations of pixel-level masks, *i.e.*, only the instance-level annotation is given for training.

Audio-visual segmentation (AVS) [1] is a recently rising problem that predicts a pixel-level map of objects that produce sound at the time of frames. A related problem to this recent task is sound source localization, which aims to locate visual regions within the frame that correspond to the sound. Sound source localization (SSL) [2, 3, 4, 5, 6, 7, 8, 9, 10] estimates a rough location of the sounding objects at a patch level, while the goal of AVS is to estimate pixel-wise segmentation masks for all sound sources. AVS requires a more comprehensive manually designed architecture than SSL since it needs to identify the exact shape of sounding objects. However, this complicated network for AVS requires countless pixel-wise accurate masks as supervision, which costs numerous labor and resources. In contrast, we aim to get rid of these costly pixel-level annotations to achieve weakly-supervised audio-visual segmentation.

Many weakly-supervised methods [11, 12, 13, 14, 15, 16, 17, 18, 19, 20] have been proposed to be successful in the visual segmentation community, where they replaced accurate pixel-level annotations with weak labels, including scribbles [13, 15], bounding boxes [11, 12], points [14], and image-level class labels [16, 17]. Typically, CAM [18] was proposed with a CNN-based image classifier to generate pseudo segmentation masks at the pixel level using coarse object localization maps. With coarse localization maps, following works tried to obtain the full extent of objects by regions expansion [21, 22, 23], stochastic inference [24], boundary constraints [25, 26, 27], or object regions mining and erasing [28, 29]. However, only one modality, *i.e.*, image is involved in these aforementioned CAM-based weakly-supervised approaches. In this work, we need to tackle with two distinctive modalities (audio and vision), where a well-designed cross-modal fusion is required to aggregate audio-visual features with high similarity in the semantic space.

37th Conference on Neural Information Processing Systems (NeurIPS 2023).

The main challenge is that we do not have pixel-level segmentation masks for this new weakly-supervised multi-modal problem during training. Without this supervision, the performance of AVS [1] on audio-visual segmentation degrades significantly, as observed in our experiments in Section 4.2. In the meanwhile, previous weakly-supervised semantic segmentation [18, 30] and visual sound source localization [9] methods have two main challenges. First, no multi-modal constraint is applied for pixel-wise mask prediction in the previous weakly-supervised semantic segmentation methods as they do not use the audio as input during training. Second, current visual sound source localization baselines generate only coarse heatmaps for sounding objects instead of binary pixel-wise segmentation masks. To address the aforementioned challenges, our key idea is to apply multi-scale multiple-instance contrastive learning in audio-visual fusion to learn multi-scale cross-modal alignment, which differs from existing weakly-supervised semantic segmentation approaches. During training, we aim to use a reliable pseudo mask with multi-scale visual features as mask-level supervision for predicting accurate audio-visual segmentation masks in a weakly-supervised setting.

To this end, we propose a novel weakly-supervised audio-visual segmentation framework, namely WS-VAS, that can predict sounding source masks from both audio and image without using the ground-truth masks. Specifically, our WS-AVS leverages multi-scale multiple-instance contrastive learning in audio-visual fusion to capture multi-scale cross-modal alignment for addressing modality uncertainty in weakly-supervised semantic segmentation methods. Then, the pseudo mask refined by contrastive class-agnostic maps will serve as pixel-level guidance during training. Compared to previous visual sound source localization approaches, our method can generate binary accurate audio-visual segmentation masks.

Empirical experiments on AVSBench [1] comprehensively demonstrate the state-of-the-art performance against previous weakly-supervised baselines. In addition, qualitative visualizations of segmentation masks vividly showcase the effectiveness of our WS-AVS in predicting sounding object masks from both audio and image. Extensive ablation studies also validate the importance of audio-visual fusion with multi-scale multiple-instance contrastive learning and pseudo mask refinement by contrastive class-agnostic maps in single-source and multi-source audio-visual segmentation.

Our main contributions can be summarized as follows:

- We investigate a new weakly-supervised multi-modal problem that predicts sounding object masks from both audio and image without needing pixel-level annotations.

- We present a novel framework for weakly-supervised audio-visual segmentation, namely WS-AVS, with multi-scale multiple-instance contrastive learning to capture multi-scale audio-visual alignment.

- Extensive experiments comprehensively demonstrate the state-of-the-art superiority of our WS-AVS over previous baselines on single-source and multi-source sounding object segmentation.

## 2 Related Work

**Audio-Visual Learning.** Audio-visual learning has been explored in many existing works [31, 32, 33, 34, 2, 35, 36, 37, 38, 39, 40, 41, 42, 43, 44, 45, 46] to learn audio-visual correspondence from those two distinct modalities in videos. Given video sequences, the goal is to push away those from different pairs while closing audio and visual representations from the same pair. Such cross-modal correspondence is useful for several tasks, such as audio/speech separation [37, 47, 48, 35, 36, 49, 50, 51, 52], visual sound source localization [2, 3, 4, 5, 6, 7, 8, 9, 10, 53, 54], audio spatialization [55, 56, 57, 38], and audio-visual parsing [58, 59, 60, 61]. In this work, we mainly focus on learning audio-visual association for pixel-level audio-visual segmentation, which is more demanding than those tasks mentioned above.

**Visual Sound Localization**. Visual sound source localization aims to identify objects or regions of a video that correspond to sounds in the video. Recent researchers [2, 4, 5, 7, 8, 62, 9, 10] used diverse networks to learn the audio-visual alignment for predicting sound sources in the video. Typically, Attention10k [2] proposed a two-stream architecture based on an attention mechanism to detect sounding objects in the image. More recently, EZVSL [9] applied a multiple-instance contrastive learning objective to learn the alignment across regions with the most corresponding audio. However, they cannot predict exact masks for sounding objects.

**Audio-Visual Segmentation.** Audio-visual segmentation is a challenging problem that predicts pixel-wise masks for sounding objects in the image. In order to address this issue, Zhou *et al.* [1] introduced the first audio-visual segmentation benchmark with pixel-level annotations for each 5-second video, where a binary mask is used to indicate the pixels of sounding objects for the corresponding audio. Moreover, they proposed an encoder-decoder network with a temporal pixel-by-pixel audio-visual interaction module to solve the pixel-level audio-visual segmentation task. However, their approach requires exhausted pixel-level annotations as the main supervision. Without this supervision, their performance degrades significantly as observed in our experiments in Section 4.2. In this work, we aim to get rid of the dependency of pixel-level labels, by leveraging only instance-level labels for weakly-supervised audio-visual segmentation.

**Weakly-Supervised Semantic Segmentation.** Weakly-supervised visual segmentation aims at reducing the burden of collecting pixel-level annotations for image segmentation at a large scale required by its fully-supervised counterpart. Early methods explored different forms of weak supervision, including image-level label [63, 64], scribble [13, 15], and bounding box [12, 65, 66]. Typically, CAM [18] was introduced to generate class activation maps as pseudo pixel-level segmentation masks for weakly-supervised object localization. Based on CAM, following works tried to optimize the coarse class activation map for more precise object localization, by expanding localization regions [21, 22, 23], using stochastic inference [24], exploring boundary constraints [25, 26, 27], or mining and erasing regions of objects [28, 29]. More recently, $C^2$AM [30] leveraged contrastive learning with the semantic relation between foreground and background as positive and negative pairs to generate a class-agnostic activation map. Different from weakly-supervised visual segmentation baselines, we aim to develop a fully novel framework to aggregate semantics from both audio and visual representations with the only instance-level annotation. To the best of our knowledge, we are the first to explore class-agnostic activation maps on weakly-supervised audio-visual segmentation. In addition, we leverage audio-visual multi-scale fusion to boost the segmentation performance.

**Weakly-Supervised Audio Learning.** Weakly-supervised learning has been widely-used in many audio-relevant tasks, such as sound events detection [67, 68, 69] and sound source separation [70, 71, 72, 73]. Typically, WAL-Net [69] proposed using weakly labeled data from the web to explore how the label density and corruption of labels affect the generalization of models. DC/GMM [71] applied an auxiliary network to generate the parameters of Gaussians in the embedding space with a one-hot vector indicating the class as input. However, the problem we need to solve in this work is more difficult and challenging. Indeed, we aim to leverage the weakly labelled supervision of image categories to solve the new weakly-supervised multi-modal problem.

# 3 Method

Given a clip of audio and video frames, our target is to predict pixel-level masks for sounding objects in the frame. We present a novel Weakly-Supervised Audio-Visual Segmentation framework named WS-AVS for single source audio-visual segmentation without involving the pixel-wise annotation, which mainly consists of two modules, Audio-Visual Fusion with Multi-scale Multiple Instance Contrastive Learning in Section 3.2 and Pseudo Mask Refinement by contrastive class-agnostic maps in Section 3.3.

## 3.1 Preliminaries

In this section, we first describe the problem setup and notations, and then revisit the first work [1] with pixel-level annotations for single source audio-visual segmentation.

**Problem Setup and Notations.** Given an audio spectrogram and an image, our goal is to predict the binary segmentation mask for the sound source in the image spatially. Let $\mathcal{D} = (a_i, v_i) : i = 1, ..., N$ be a dataset of paired audio $a_i \in \mathbb{R}^{T \times F}$ and visual data $v_i \in \mathbb{R}^{3 \times H \times W}$, where only one sound source $a_i$ are assumed to be existing in $v_i$. Note that $T, F$ denotes the dimension of time and frequency of the audio spectrogram. We follow previous work [1, 9] and first encode the audio and visual inputs using a two-stream neural network encoder and projection heads, denoted as $f_a(\cdot), g_a(\cdot)$ and $f_v(\cdot), g_v(\cdot)$ for the audio and images, separately. The audio encoder computes global audio representations $\mathbf{a}_i = g_a(f_a(a_i)), \mathbf{a}_i \in \mathbb{R}^{1 \times D}$ and the visual encoder extracts multi-scale representations $\{\mathbf{v}_i^s\}_{s=1}^{S} = g_v(f_v(v_i)), \mathbf{v}_i^s \in \mathbb{R}^{D \times H^s \times W^s}$ for each $s$th stage. During the training, we do not have mask-level annotations $\mathbf{Y} \in \mathbb{R}^{H \times W}$.

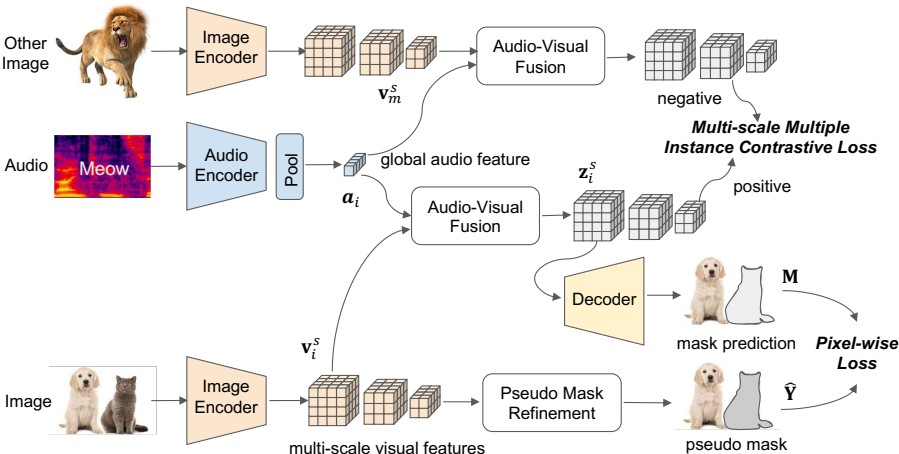

Figure 1: Illustration of the proposed Weakly-Supervised Audio-Visual Segmentation (WS-AVS) framework. Global audio features $\mathbf{a}_i$ and multi-scale visual features $\{\mathbf{v}_i^s\}_{s=1}^S$ are extracted from the audio spectrogram and image via an audio encoder and an image encoder, respectively. The Audio-Visual Fusion module combines multi-scale multiple-instance contrastive learning to capture multi-scale cross-modal alignment between global audio and multi-scale visual features. Then, with multi-scale visual features $\{\mathbf{v}_i^s\}_{s=1}^S$, the Pseudo Mask Refinement module generates a reliable pseudo mask $\hat{\mathbf{Y}}$ for sounding objects by using contrast class-agnostic map. Finally, a decoder is applied on multi-scale audio-visual features $\{\mathbf{z}_i^s\}_{s=1}^S$ to predict masks. The pixel-wise loss is simply optimized between the predicted mask $\mathbf{M}$ and the pseudo mask $\hat{\mathbf{Y}}$.

**Revisit Audio-Visual Segmentation with Pixel-level Annotations.** To solve the single source audio-visual segmentation problem, AVS [1] introduced the pixel-wise audio-visual fusion module to encode the multi-scale visual features and global audio representations. After the cross-modal fusion, the audio-visual feature map $\mathbf{z}_i^s$ at the $s$th stage is updated as:

$$\mathbf{z}_i^s = \mathbf{v}_i^s + \mu\left(\frac{\theta(\mathbf{v}_i^s)\phi(\hat{\mathbf{a}}_i)^\top}{H^s W^s}\omega(\mathbf{v}_i^s)\right) \tag{1}$$

where $\hat{\mathbf{a}}_i \in \mathbb{R}^{D \times H^s \times W^s}$ denotes the duplicated version of the global audio representation $\mathbf{a}_i$ that repeats $H^s W^s$ times at the $s$th stage. $\mu(\cdot), \theta(\cdot), \phi(\cdot), \omega(\cdot)$ denote the $1 \times 1 \times 1$ convolution operator. Then those updated multi-stage feature maps are passed into the decoder of Panoptic-FPN [74] and a sigmoid activation layer to generate the final output mask $\mathbf{M} \in \mathbb{R}^{H \times W}$. With the pixel-level annotation $\mathbf{Y}$ as supervision, they applied the binary cross entropy loss between the prediction and label as:

$$\mathcal{L}_{\text{baseline}} = \text{BCE}(\mathbf{M}, \mathbf{Y}) \tag{2}$$

However, such a training mechanism is extremely dependent on pixel-level annotations. Without this supervision, their performance on single source audio-visual segmentation deteriorates significantly, as observed in Section 4.2. In the meanwhile, current weakly-supervised semantic segmentation (WSSS) [18, 30] and visual sound source localization (VSSL) [9] methods will pose two main challenges. First, there is no multi-modal constraint for pixel-level mask prediction in the previous WSSS methods as they do not involve the audio during training. Second, existing VSSL approaches only predict coarse heatmaps for sounding sources instead of accurate pixel-wise segmentation masks. To address these challenges, we propose a novel weakly-supervised audio-visual segmentation framework for single sounding object segmentation, which is composed of Audio-Visual Fusion with multi-scale multiple-instance contrastive learning and Pseudo Mask Refinement by contrastive class-agnostic map, as shown in Figure 1.

### 3.2 Audio-Visual Fusion with Multi-scale Multiple-Instance Contrastive Learning

In order to address the modality uncertainty brought by the previous WSSS baselines [18, 30], inspired by EZ-VSL [9], we propose to conduct audio-visual fusion by focusing only on the most

aligned regions when matching the audio to the multi-scale visual feature. This is because most locations in the video frame do not correspond to the sounding object, and thus the representations at these locations should not be aligned with the audio during training.

To align the audio and visual features at locations corresponding to sounding sources, we apply the multi-scale multiple-instance contrastive learning (M$^2$ICL) objective to align at least one location in the corresponding bag of multi-scale visual features with the audio representation in the same mini-batch, which is defined as:

$$\mathcal{L}_{a \to v} = -\frac{1}{B} \sum_{i=1}^{B} \sum_{s=1}^{S} \log \frac{\exp\left(\frac{1}{\tau} \mathtt{sim}(\mathbf{a}_i, \mathbf{v}_i^s)\right)}{\sum_{m=1}^{B} \exp\left(\frac{1}{\tau} \mathtt{sim}(\mathbf{a}_i, \mathbf{v}_m^s)\right)} \tag{3}$$

where the similarity $\mathtt{sim}(\mathbf{a}_i, \mathbf{v}^s)$ denotes the max-pooled audio-visual cosine similarity of $\mathbf{a}_i$ and $\mathbf{v}_i^s$ across all spatial locations at $s$ stage. That is, $\mathtt{sim}(\mathbf{a}_i, \mathbf{v}_i^s) = \max_{H^s W^s}(\mathbf{a}_i, \mathbf{v}_i^s)$. $B$ is the batch size, $D$ is the dimension size, and $\tau$ is a temperature hyper-parameter.

Similar to EZ-VSL [9], we use a symmetric loss of Eq. 4 to discriminate negative audio bags from other audio samples in the same mini-batch, which is defined as

$$\mathcal{L}_{v \to a} = -\frac{1}{B} \sum_{i=1}^{B} \sum_{s=1}^{S} \log \frac{\exp\left(\frac{1}{\tau} \mathtt{sim}(\mathbf{a}_i, \mathbf{v}_i^s)\right)}{\sum_{m=1}^{B} \exp\left(\frac{1}{\tau} \mathtt{sim}(\mathbf{a}_m, \mathbf{v}_i^s)\right)} \tag{4}$$

where $\mathbf{a}_m$ denote the global audio visual from other sample $m$ in the mini-batch. The overall audio-visual fusion objective with the multi-scale multiple-instance contrastive learning mechanism is defined as:

$$\mathcal{L}_{\mathrm{avf}} = -\frac{1}{B} \sum_{i=1}^{B} \sum_{s=1}^{S} \log \frac{\exp\left(\frac{1}{\tau} \mathtt{sim}(\mathbf{a}_i, \mathbf{v}_i^s)\right)}{\sum_{m=1}^{B} \exp\left(\frac{1}{\tau} \mathtt{sim}(\mathbf{a}_i, \mathbf{v}_m^s)\right)} + \log \frac{\exp\left(\frac{1}{\tau} \mathtt{sim}(\mathbf{a}_i, \mathbf{v}_i^s)\right)}{\sum_{m=1}^{B} \exp\left(\frac{1}{\tau} \mathtt{sim}(\mathbf{a}_m, \mathbf{v}_i^s)\right)} \tag{5}$$

Optimizing the loss will push the model to learn discriminatively global audio representation $\mathbf{a}_i$ and multi-scale visual features $\{\mathbf{v}_i^s\}_{s=1}^{S}$. Then, these features are fed forward into Eq. 1 to generate updated multi-scale audio-visual features $\{\mathbf{z}_i^s\}_{s=1}^{S}$. Finally, with these updated multi-scale audio-visual features, we apply the decoder of Panoptic-FPN [74] to generate the output mask $\mathbf{M}$. Note that, the recent audio-visual segmentation baseline [1] does not give any cross-modal constraint on the audio and visual representations in the audio-visual fusion, which causes a significant performance drop when removing the ground-truth masks during training.

### 3.3 Pseudo Mask Refinement by Contrastive Class-agnostic Maps

The second challenge of spatial uncertainty in VSSL approaches [9] requires us to utilize mask-level supervision for generating precise final output mask $\mathbf{M}$. To generate reliable pseudo masks of training sets, motivated by recent WSSL pipelines [30], we introduce the Pseudo Mask Refinement by applying a contrastive class-agnostic map on multi-scale visual features $\{\mathbf{v}_i^s\}_{s=1}^{S}$. Specifically, we utilize the contrastive loss in [30] to close the distance between the representations in positive pairs (foreground-foreground, background-background) and push away the representations in negative pairs (foreground-background). Then, we use the background activation maps as pseudo labels to further train a salient object detector [75] to predict the salient region $\mathbf{S} \in \mathbb{R}^{1 \times H \times W}$ in the image. Finally, the predicted salient cues are concatenated with the initial class-agnostic map $\mathbf{A} \in \mathbb{R}^{D \times H \times W}$ to perform the argmax operator along the embedding dimension to generate the binary pseudo mask $\hat{\mathbf{Y}} \in \mathbb{R}^{H \times W}$, which is formulated as:

$$\hat{\mathbf{Y}} = \mathbf{L}\left[\arg\max_{1+D}([\mathbf{S}; \mathbf{A}])\right] \tag{6}$$

where $[\;;\;]$ denotes the concatenation operator. Note that since we are focusing on single-source segmentation, $\mathbf{L} = [\mathbf{L}_S; \mathbf{L}_A] \in \mathbb{R}^{(1+D) \times H \times W}$. $\mathbf{L}_S \in \mathbb{R}^{1 \times H \times W}$ is the full label of zeros for the salient cues, while $\mathbf{L}_A \in \mathbb{R}^{D \times H \times W}$ is the full label of ones for the class-agnostic maps. With the

Table 1: Comparison results (%) of weakly-supervised audio-visual segmentation. "ws" denotes the weakly-supervised baseline, where only the instance-level class is used for training.

| Method | Single Source | | Multiple Source | |
|---|---|---|---|---|
| | mIoU (↑) | F-score (↑) | mIoU (↑) | F-score (↑) |
| AVS [1] (ws) | 12.63 | 24.99 | 8.76 | 15.72 |
| CAM [18] | 19.26 | 27.88 | 12.65 | 19.83 |
| EZ-VSL [9] | 29.40 | 35.70 | 23.58 | 27.31 |
| $C^2$AM [30] | 30.87 | 36.55 | 25.33 | 29.58 |
| WS-AVS (ours) | **34.13** (**+3.26**) | **51.76** (**+15.21**) | **30.85** (**+5.52**) | 46.87 (**+17.29**) |

reliable pseudo mask refinement, we apply the binary cross entropy loss between the predicted mask $\mathbf{M}$ and the pseudo mask $\hat{\mathbf{Y}}$, which is defined as:

$$\mathcal{L}_{\text{pmr}} = \text{BCE}(\mathbf{M}, \hat{\mathbf{Y}}) \tag{7}$$

The overall objective of our model is simply optimized in an end-to-end manner as:

$$\mathcal{L} = \mathcal{L}_{\text{avf}} + \mathcal{L}_{\text{pmr}} \tag{8}$$

During inference, we follow previous work [1] and directly use the predicted mask $\mathbf{M} \in \mathbb{R}^{H \times W}$ as the final output. It is worth noting that the final localization map in the current VSSL methods [9] was generated through bilinear interpolation of the audio-visual feature map at the last stage.

## 4 Experiments

### 4.1 Experiment Setup

**Datasets.** AVSBench [1] contains 4,932 videos with 10,852 total frames from 23 categories including animals, humans, instruments, etc. Following prior work [1], we use the split of 3,452/740/740 videos for train/val/test in single source segmentation.

**Evaluation Metrics.** Following previous work [1], we use the averaged IoU (mIoU) and F-score to evaluate the audio-visual segmentation performance. mIoU computes the intersection-over-union (IoU) of the predicted mask and ground-truth mask for evaluating the region similarity. F-score calculates both the precision and recall for evaluating the contour accuracy.

**Implementation.** The input image is resized to a resolution of $224 \times 224$. The input audio takes the log spectrograms extracted from 3s of audio at a sample rate of 22050Hz. Following previous works [9, 10], we apply STFT to generate an input tensor of size $257 \times 300$ (257 frequency bands over 300 timesteps) using 50ms windows with a hop size of 25ms. We follow the prior audio-visual segmentation work [1] and use the ResNet50 [76] as the audio and visual encoder. The visual model is initialized using weights pre-trained on ImageNet [77]. The model is trained with the Adam optimizer with default hyper-parameters $\beta_1 = 0.9$, $\beta_2 = 0.999$, and a learning rate of 1e-4. The model is trained for 20 epochs with a batch size of 64.

### 4.2 Comparison to Prior Work

In this work, we propose a novel and effective framework for weakly-supervised audio-visual segmentation. To validate the effectiveness of the proposed WS-AVS, we comprehensively compare it to previous audio-visual segmentation, weakly-supervised semantic segmentation, and visual sound source localization baselines: 1) AVS [1](ws): the weakly-supervised version of the first audio-visual segmentation work, where we removed the pixel-level annotations for training; 2) CAM [18]: the first baseline using class-activation maps as pseudo segmentation masks for weakly-supervised object localization; 3) EZ-VSL [9]: the state-of-the-art visual sound source localization approach with coarse source maps as output; 4) $C^2$AM [30]: the very recent work for weakly-supervised semantic segmentation with only the image as input.

The quantitative comparison results are reported in Table 1. As can be seen, we achieve the best performance in terms of all metrics compared to previous weakly-supervised baselines. In particular,

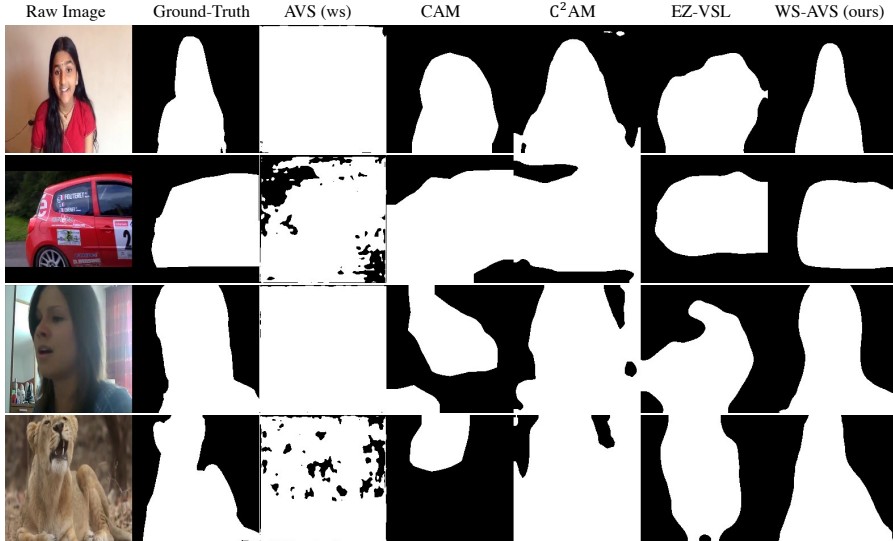

| Raw Image | Ground-Truth | AVS (ws) | CAM | C$^2$AM | EZ-VSL | WS-AVS (ours) |

Figure 2: Qualitative comparisons with weakly-supervised audio-visual segmentation (AVS [1]), weakly-supervised semantic segmentation (CAM [18], C$^2$AM [30]), and visual sound source localization (EZ-VSL [9]) baselines on single-source segmentation. The proposed WS-AVS produces much more accurate and high-quality segmentation maps for sounding objects.

the proposed WS-AVS significantly outperforms AVS [1](ws), the weakly-supervised version of the state-of-the-art audio-visual segmentation approach, by 21.50 mIoU and 26.77 F-score. Moreover, we achieve superior performance gains of 3.26 mIoU and 15.21 F-score compared to C$^2$AM [30], which implies the importance of the proposed multi-scale multiple-instance contrastive learning for addressing the modality uncertainty in audio-visual fusion. Meanwhile, our WS-AVS outperforms EZ-VSL [9], the current state-of-the-art visual sound source localization baseline, by a large margin, where we achieve the performance gains of 4.73 mIoU and 16.06 F-score. These significant improvements demonstrate the superiority of our method in single-source audio-visual segmentation.

In addition, significant gains in multi-source sound localization can be observed in Table 1. Compared to AVS [1](ws), the weakly-supervised version of the state-of-the-art audio-visual segmentation approach, we achieve the results gains of 22.09 mIoU and 31.15 F1 score. Furthermore, the proposed approach still outperforms C$^2$AM [30] by 5.52 mIoU and 17.29 F1 score. We also achieve highly better results than EZ-VSL [9], the current state-of-the-art visual sound source localization baseline. These results validate the effectiveness of our approach in learning multi-scale multiple-instance semantics from mixtures and images for multi-source localization.

In order to qualitatively evaluate the localization maps, we compare the proposed WS-AVS with weakly-supervised AVS [1], CAM [18], C$^2$AM [30], and EZ-VSL [9] on single-source segmentation in Figure 2. From comparisons, three main observations can be derived: 1) Without pixel-level annotations, AVS [1], the first audio-visual segmentation fails to predict the masks for sounding objects; 2) the quality of segmentation masks generated by our method is much better than the strong visual sound source baseline, EZ-VSL [9]; 3) the proposed WS-AVS achieves competitive even better results on predicted masks against the weakly-supervised semantic segmentation baseline [30] by using contrast class-agnostic maps for prediction. These visualizations further showcase the superiority of our simple WS-AVS with multi-scale multiple-instance contrastive learning to guide segmentation for predicting accurate source masks.

### 4.3 Experimental Analysis

In this section, we performed ablation studies to demonstrate the benefit of introducing the Audio-Visual Fusion with multi-scale multiple-instance contrastive learning and Pseudo Mask Refinement module by contrastive class-agnostic map. We also conducted extensive experiments to explore the effect of fusion stages and batch size on weakly-supervised audio-visual segmentation. Furthermore,

Table 2: Ablation studies on Audio-Visual Fusion (AVF) and Pseudo Mask Refinement (PMR).

| AVF | PMR | mIoU (↑) | F-score (↑) |
|-----|-----|----------|-------------|
| ✗ | ✗ | 12.63 | 24.99 |
| ✓ | ✗ | 31.85 (**+19.22**) | 39.78 (**+14.79**) |
| ✗ | ✓ | 32.46 (**+19.83**) | 43.92 (**+18.93**) |
| ✓ | ✓ | **34.13** (**+21.50**) | **51.76** (**+26.77**) |

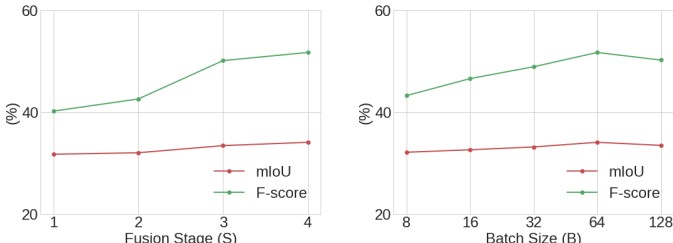

Figure 3: Effect of fusion stages and batch size on mIoU and F-score.

we visualized high-quality pseudo masks from Pseudo Mask Refinement in weakly-supervised training.

**Audio-Visual Fusion & Pseudo Mask Refinement.** To validate the effectiveness of the introduced audio-visual fusion with multi-scale multiple-instance contrastive learning (AVF) and Pseudo Mask Refinement by contrastive class-agnostic map (PMR), we ablate the necessity of each module and report the quantitative results in Table 2. We can observe that adding AVF to the vanilla baseline highly increases the results of single-source audio-visual segmentation by 19.22 mIoU and 14.79 F-score, which demonstrates the benefit of multi-scale multiple-instance contrastive learning ($M^2ICL$) in extracting aligned cross-modal feature in fusion for source segmentation. Meanwhile, introducing only PMR in the baseline also increases the segmentation performance in terms of all metrics. More importantly, incorporating AVF with $M^2ICL$ and PMR together into the baseline significantly raises the performance by 21.50 mIoU and 26.77 F-score. These improving results validate the importance of AVF with $M^2ICL$ and PMR with contrastive class-agnostic map in addressing both modality and spatial challenges for generating accurate audio-visua masks.

**Effect of Fusion Stages and Batch Size.** The number of fusion stages and batch size used in the proposed $M^2ICL$ affect the extracted cross-modal representations for audio-visual segmentation. To explore such effects more comprehensively, we varied the number of fusion stages from $\{1, 2, 3, 4\}$ and ablated the batch size from $\{8, 16, 32, 64, 128\}$. The comparison results of segmentation performance are shown in Figure 3. When the number of fusion stages is 4 and using batch size of 64 in $M^2ICL$, we achieve the best segmentation performance in terms of all metrics. With the increase of fusion stages from 1 to 4, the proposed WS-AVS consistently raises results, which shows the importance of multi-scale visual features in audio-visual fusion for learning discriminative cross-modal representations. Regarding the batch size, the performance of the proposed WS-AVS climbs with the increase of the batch size from 8 to 64. However, increasing the batch size from 64 to 128 will not continually improve the result since there might be some false negatives in the mini-batch for this training set with a relatively smaller size.

**Generated Pseudo Mask.** Generating reliable pseudo masks with contrast class-agnostic maps is critical for us to train the weakly-supervised framework. To better evaluate the quality of generated pseudo masks, we visualize the pseudo mask and class-agnostic map in Figure 4. As can be observed in the last column, the class-agnostic map can successfully localize the sounding object in the image. Furthermore, the pseudo masks in the third column predicted by Eq. 6 have high quality and they are even fine-grained compared to the manually annotated ground-truth masks. These meaningful visualization results further showcase the success of the Pseudo Mask Refinement in extracting reliable pseudo masks as guidance for single-source audio-visual segmentation.

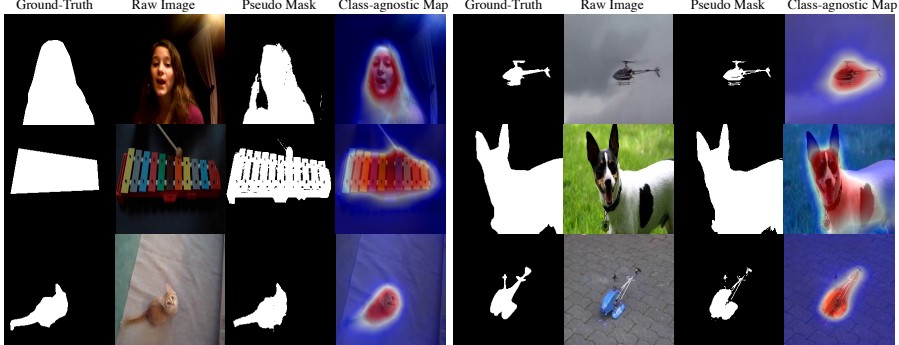

Figure 4: Qualitative comparisons of the ground-truth mask, pseudo mask, and class-agnostic map. Note that the generated pseudo mask is used for weakly-supervised training.

## 4.4 Limitation

Although the proposed WS-AVS achieves superior results on single-source audio-visual segmentation, the performance gains of our approach on the mIoU metric are not significant. One possible reason is that our model easily overfits across the training phase, and the solution is to incorporate dropout and momentum encoders together for weakly-supervised audio-visual segmentation. Meanwhile, we notice that our model performs worse on mixed sound sources, such as a scenario with both female singing and playing tabla. The future work could be to add separation objectives or assign semantic labels to both audio and visual segments in audible videos.

## 5 Conclusion

In this work, we successfully investigate the weakly-supervised audio-visual segmentation to eliminate pixel-level annotations as supervision. Then we present WS-AVS, a novel Weakly-Supervised Audio-Visual Segmentation framework with multi-scale multiple-instance contrastive learning for capturing multi-scale audio-visual alignment. Furthermore, we conduct extensive experiments on AVSBench to demonstrate the superiority of our WS-AVS against previous weakly-supervised audio-visual segmentation, weakly-supervised semantic segmentation, and visual sound source localization approaches.

**Broader Impact.** The proposed approach successfully predicts segmentation masks of sounding sources from manually-collected datasets on the web, which might cause the model to learn internal biases in the data. For instance, the model could fail to discover rare but crucial sound sources. Therefore, these issues should be addressed for the deployment of real applications.

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
