# OpenReview forum: "Weakly-Supervised Audio-Visual Segmentation"
_NeurIPS.cc/2023/Conference — NeurIPS 2023 poster_

### Official Review · Reviewer_XL4i · 2023-07-04

**Soundness:** 3 good
**Presentation:** 3 good
**Contribution:** 3 good
**Rating:** 6
**Confidence:** 4

**Summary:**

This paper presents a new framework for weakly supervised audio-visual segmentation which does not need pixel level annotations. This is achieved via the proposed multi-scale multiple-instance contrastive learning approach which can capture audio-visual alignment in multiple scales. Comparison with existing methods show that the proposed methodology leads to state-of-the-art results.

**Strengths:**

The paper proposes an interesting approach ti audio-visual segmentation which does not rely on pixel-level annotations. This is an important contribution since obtaining such annotations is time consuming.

The paper is easy to follow.

Convincing results are presented and a detailed ablation study is presented.

**Weaknesses:**

The main weakness of the paper is that several important details are not described.

- In table 1, results for multiple sources are presented without providing details, e.g., how many sources are used, how were these examples obtained?

- It is not clear how the multi-scale features are generated.

- How are the negative examples selected in multi-scale multiple instance contrastive learning? Would be very helpful if the authors elaborate on this.

- Equal weights are used for the two loss terms in Eq. 8. Have the authors considered using different weights? This might improve the performance.

- It is not clear why the face and body are predicted as the sound producing regions in case a person speaks. Ideally, just the face or just the mouth should be identified as the sound source.

- The audio which corresponds to the images in Fig.2 is not presented. Most of the examples are included in the supplementary material so adding such a comment in the caption would be helpful.

- Finally, it would be helpful to add additional failure cases. Only one example is included in the supplementary material.

Another weakness, is that only one dataset is used for evaluation.

**Questions:**

Please see the questions above.

**Limitations:**

As mentioned above, it would be desirable if the authors include additional failure cases in the supplementary material. At the moment there is only one.

---

> ### Author Rebuttal · Authors · 2023-08-09
>
> Dear Reviewer XL4i,
>
> Thank you for appreciating our approach. We address your comments below.
>
>
> > Details on multiple sources.
>
> We use two or three sound sources from 2,120 frames for evaluation. For pseudo-labels for multi-source scenarios, we use the background activation maps as pseudo-labels and categories of each source to further train a salient object [67]. Then generated pseudo-mask separately for each sound source according to Equation (6) by replacing the salient regions with class-specific salient regions $S^\prime\in\mathbb{R}^{M\times H\times W}. Finally, we apply the loss defined in Equation (7) on each source. We will add this clarification to the revision.
>
>
> > Clarification on multi-scale features.
>
> The multi-scale features with sizes of {56x56, 28x28, 14x14, 7x7} are generated from four stages ([3,4,6,3]) of the ResNet50-based image encoder.
>
>
> > Negative examples.
>
> Following the spirit of contrastive learning methods, negative examples in multi-scale multiple instance contrastive learning are obtained from other images in the same mini-batch.
>
>
> > Different weights.
>
> Yes. We have explored different weights but did not see significant improvements (0.1%) in the performance.
>
>
> > Clarification on the prediction of face and body as the sound producing regions.
>
> This new weakly-supervised AVS problem is challenging to generate a fine-grained sound source from a small object, such as the mouth. Meanwhile, many objects with diverse semantic might be included in the video frames. Even for the ground-truth, the face and body are also annotated as the sounding sources when a person speaks or sings.
>
>
> > Adding an audio comment.
>
> Thanks for the suggestion. We will add the comment to the caption in the updated version.
>
> > More failure cases.
>
> The model may fail when two sources have very similar visual semantics, e.g., a mixture of people cheering and people crowd. We will add more failure cases to the supplementary.
>
>
> > Evaluation of more datasets.
>
> This is a great suggestion! We extended our method to experiments on Flickr-SoundNet and VGG Sound Source, and reported the comparison results of CIoU with existing approaches in the Table below. Compared to previous methods, our WS-AVS achieves the best results in both benchmarks.
>
> | Method        | Flickr-SoundNet | VGG Sound Source |
> | ---- | :----: | :----: |
> | DSOL [a]      | 74.00           | 29.91            |
> | EZVSL [b]     | 83.94           | 38.85            |
> | SLAVC [c]     | 86.40           | 39.80            |
> | WS-AVS (ours) | **91.63**           | **42.72**            |
>
>
> To verify the robustness of our proposed approach to off-screen/silent objects, we evaluated our WS-AVS on Extended Flickr-SoundNet and  Extended VGG-SS proposed in [c]. The comparison results of max-F1 with previous methods are reported in the Table below. Compared to the state-of-the-art frameworks, our WS-AVS achieves the best performance in both datasets.
>
>
> | Method        | Extended Flickr-SoundNet | Extended VGG Sound Source |
> | ---- | :----: | :----: |
> | DSOL [a]      | 49.40                    | 25.60                     |
> | EZVSL [b]     | 54.60                    | 30.90                     |
> | SLAVC [c]     | 60.10                    | 41.50                     |
> | WS-AVS (ours) | **65.27**                    | **52.38**                     |
>
>
> **References**
>
> [a] Hu, et al. Discriminative Sounding Objects Localization via Self-supervised Audiovisual Matching. In NeurIPS, 2020.
>
> [b] Mo, et al. Localizing Visual Sounds the Easy Way. In ECCV, 2022.
>
> [c] Mo, et al. A Closer Look at Weakly-Supervised Audio-Visual Source Localization. In NeurIPS, 2022.

---

> > ### Comment · Reviewer_XL4i · 2023-08-20
> >
> > The authors have addressed my concerns. Please make sure that you include the additional results and explanations in the paper.

---

> > ### Comment · Reviewer_XL4i · 2023-08-20
> >
> > The authors have addressed my concerns. Please make sure that you include the additional results and explanations in the paper.

---

> > > ### Author Response · Authors · 2023-08-20
> > > **Response to the reviewer**
> > >
> > > Dear Reviewer XL4i,
> > >
> > > Thank you for your response and we appreciate your support. We will incorporate the additional results and explanations you mentioned into the revised version of the paper. We believe that these additions will further strengthen the paper and provide a more comprehensive evaluation of our method.
> > >
> > > Thank you once again for your valuable feedback and for your continued support throughout the review process.

---

### Official Review · Reviewer_DJ6f · 2023-07-04

**Soundness:** 3 good
**Presentation:** 3 good
**Contribution:** 2 fair
**Rating:** 5
**Confidence:** 4

**Summary:**

This work introduces a new setting for audio-visual segmentation, Weakly-Supervised Audio-Visual Segmentation (WS-AVS). The authors address the challenge of the costly and not always available pixel-level masks by employing weakly-supervised audio-visual segmentation. This framework uses multi-scale multiple-instance contrastive learning for  capturing multi-scale audio-visual alignment. Tested on the AVS Bench, WS-AVS demonstrates superior performance in both single-source and multi-source scenarios when compared to previous methods.


**Strengths:**

1. To eliminate the requirement of pixel-level annotations, this work propose to solve the audio-visual segmentation problem in a weakly-supervised way. Correspondingly, it uses adopts Audio-Visual Fusion with Multi-scale Multiple-Instance Contrastive Learning and Pseudo Mask Refinement by Contrastive Class-agnostic Maps to solve the challenges in the weakly supervised setting.


2. The authors provide comprehensive evaluation under various setting to illustrate the performance of the proposed method.

**Weaknesses:**

1. Clarifying Novelty
The paper's novelty looks somewhat unclear. This reviewer recommends clearly defining the unique technical contributions proposed by this work, especially those set apart from the developments in weakly-supervised segmentation. For instance, it would be advantageous to underscore the importance of Multi-scale Multiple-Instance Contrastive Learning in the given task and its distinctiveness from similar concepts previously employed in weakly-supervised segmentation techniques.

2. Further analysis
The authors offer sufficient ablation studies to ascertain the performance improvement attributed to AVF and PMR. However, the reviewer anticipates a more in-depth analysis of the outcomes. Specifically, what distinguishes the audio-visual relationship as learned via supervised methods from those learned in weakly-supervised ways? Considering the final classification outcome, it's intuitive to hypothesize that weakly-supervised methods might result in a sub-optimal or noisy audio-visual mapping. But, without human bias, these weakly-supervised maps might demonstrate superior generalization or aid in identifying human annotation-induced preconceptions. The reviewer encourages exploring these insights rather than viewing weakly-supervised audio-visual segmentation merely as a workaround for data limitations.



**Questions:**

The reviewer believes the task of weakly-supervised audio-visual segmentation is worth exploring, and hence currently vote for borderline accept. However, the authors should adequately discuss the key novelty of this paper for the final acceptance.

---

> ### Author Rebuttal · Authors · 2023-08-09
>
> Dear Reviewer DJ6f,
>
> Thank you for appreciating our approach. We address your comments below.
>
>
> > Clarification on contributions/novelty.
>
> Our paper includes two main technical contributions:
>
> 1) We are the first to investigate a new weakly-supervised multi-modal problem that predicts sounding object masks from both audio and image without needing pixel-level annotations.
>
> 2) We present a simple but effective framework for weakly-supervised audio-visual segmentation by combining multi-scale multiple-instance contrastive learning and pseudo mask refinement to achieve state-of-the-art results on single-source and multi-source sounding object segmentation.
>
> Based on the new problem and effective method, we believe that our paper has great potential to become a good benchmark approach for weakly supervised audio-visual segmentation.
>
>
>
> > Further analysis on audio-visual matching in supervised/weakly-supervised AVS.
>
> This is a good suggestion! To explore the effect of audio-visual matching loss introduced in supervised AVS [1] for multi-source segmentations, we added the audio-visual matching loss to both the baseline and our WS-AVS, and reported the comparison results on multi-source segmentations under weakly-supervised AVS setting in the Table below. Applying this audio-visual matching loss benefits the audio-visual fusion for weakly-supervised AVS in terms of all metrics. Besides, the magnitude of improvement is large than 1% reported in supervised AVS [1], which indicates that the audio-visual relationship learned in weakly-supervised AVS is more important than that learned from supervised settings.
>
>
> | Method        | AVM     | mIoU (↑)  | F-score (↑)  |
> | ---- | :----: | :----: | :----: |
> | AVS           | &cross; | 8.76      | 15.72        |
> | AVS           | &check; | **9.85**     | **17.36**        |
> | WS-AVS (ours) | &cross; | 30.85     | 46.87        |
> | WS-AVS (ours) | &check; | **32.03**     | **49.15**        |

---

> > ### Comment · Reviewer_DJ6f · 2023-08-16
> >
> > After reading the rebuttal and the comments from the other reviewers, I tend to keep my current score. Thanks for addressing the concern and conducting the experiments.

---

### Official Review · Reviewer_yiGS · 2023-07-05

**Soundness:** 3 good
**Presentation:** 3 good
**Contribution:** 2 fair
**Rating:** 6
**Confidence:** 4

**Summary:**

This paper proposes a new contrastive learning approach with pseudo-mask generation/refinement process for weakly supervised video segmentation with audio guidance. The authors show that their method outperforms several other state-of-the-art methods for the visual segmentation task when training and testing on the AVSBench dataset.


**Strengths:**

- The paper shows strong results compared to other methods in the literature.
- The paper tries to tackle a very important problem in the vicinity of audio-visual perception which is how to find the pixel regions that correspond to sounding sources when limited annotated data are available.
- The paper is well written and the ideas are clearly presented.


**Weaknesses:**

Unfortunately the paper also contains several weaknesses that narrow down the scope and the potential impact of the paper that could be addressed using experiments and rewriting some parts of the manuscript when needed.

- The authors only include experiments with a very small scale in terms of multiple aspects (video frames, classes of sounds only 23 objects, number of videos < 5k) to compare their method with other established methods. Although having smaller scale experiments help with finetuning of the hyperparameters and early experimentation, it becomes difficult to extrapolate the findings of this study to large scale solutions and real-world data with only showing results on the chosen dataset. Some dataset that come on top of my head and could be used by the authors to enrich the results and show the true potential of their method would be some more limited datasets like Flickr-SoundNet [A], a medium-sized mostly single-source dataset like VGG Sound Source [B] or a real-world video dataset like AudioSet [C] or YFCC100M [D].
- Building upon my previous argument, using more realistic and large scale datasets would not only put away potential skepticism revolving around the validity of training neural networks with only 5k examples and limited video resolutions (e.g. capturing motion features when the average video has 2 frames is almost impossible for the dataset used in this paper) but would also show whether the proposed approach is robust when trained with videos that contain no on-screen objects which is very difficult to solve for related problems like on-screen sound separation [E]. I don’t want to extend the problem to cases where more than one sounds appear on-screen / off-screen and thus some audio separation front-end might be needed before obtaining the audio features but I would still expect some more convincing large-scale experiments.
- In the current version of the paper, the authors only consider evaluation datasets where the sounding object appears on-screen at some point in the video but they completely omit to report the performance for off-screen-only videos. To that end, the authors should also evaluate the performance of their model when there is some visual activity on-screen but the sounding object/action cannot be seen in camera and test how robust their model is. According to my experience, there will be a trade-off in the sensitivity of how accurate the model will predict the masks for on-screen objects which correspond to the on-screen audio and how well the model will predict zeros for on-screen objects and actions that do not make any sound (e.g. a human talking in the background while a non-talking human is displayed on-screen).

Overall, I am willing to increase my score if most of the most important above concerns are addressed by the authors since I truly believe that the paper has a great potential to become a good benchmark method for weakly supervised audio-guided video segmentation.

[A] Arda Senocak, Tae-Hyun Oh, Junsik Kim, Ming-Hsuan Yang, and In So Kweon. Learning to localize sound source in visual scenes. In Proceedings of the IEEE Conference on Computer Vision and Pattern Recognition, pages 4358–4366, 2018.

[B] Honglie Chen, Weidi Xie, Andrea Vedaldi, and Andrew Zisserman. Vggsound: A large-scale audio-visual dataset. In International Conference on Acoustics, Speech, and Signal Processing (ICASSP), 2020.

[C] Gemmeke, J.F., Ellis, D.P., Freedman, D., Jansen, A., Lawrence, W., Moore, R.C., Plakal, M. and Ritter, M., 2017, March. Audio set: An ontology and human-labeled dataset for audio events. In 2017 IEEE international conference on acoustics, speech and signal processing (ICASSP) (pp. 776-780). IEEE.

[D] Thomee, B., Shamma, D.A., Friedland, G., Elizalde, B., Ni, K., Poland, D., Borth, D. and Li, L.J., 2016. YFCC100M: The new data in multimedia research. Communications of the ACM, 59(2), pp.64-73.

[E] Tzinis E, Wisdom S, Jansen A, Hershey S, Remez T, Ellis D, Hershey JR. Into the Wild with AudioScope: Unsupervised Audio-Visual Separation of On-Screen Sounds. In International Conference on Learning Representations 2021.


**Questions:**

Have you considered an analysis of the proposed approach for the choice of the model for ensuring robustness towards the “reliable pseudo mask extraction with multi-scale visual features”?


**Limitations:**

The authors discuss some limitations of their work and I think they should include the issues that I raised in the Section above and that they will not be able to address through experiments and/or rebuttal.

---

> ### Author Rebuttal · Authors · 2023-08-09
>
> Dear Reviewer yiGS,
>
> Thank you for the detailed review. We will address your concerns below.
>
>
> > Large-scale experiments.
>
> This is a great suggestion! We extended our method to experiments on Flickr-SoundNet and VGG Sound Source, and reported the comparison results of CIoU with existing approaches in the Table below. Compared to previous methods, our WS-AVS achieves the best results in both benchmarks.
>
> | Method        | Flickr-SoundNet | VGG Sound Source |
> | ---- | :----: | :----: |
> | DSOL [a]      | 74.00           | 29.91            |
> | EZVSL [b]     | 83.94           | 38.85            |
> | SLAVC [c]     | 86.40           | 39.80            |
> | WS-AVS (ours) | **91.63**           | **42.72**            |
>
> > Robustness to non-sounding and off-screen objects.
>
> This is a good suggestion! To verify the robustness of our proposed approach to non-sounding and off-screen objects, we evaluated our WS-AVS on Extended Flickr-SoundNet and  Extended VGG-SS proposed in [c]. The comparison results of max-F1 with previous methods are shown in the Table below. Compared to the state-of-the-art frameworks, our WS-AVS achieves the best performance in both datasets.
>
> | Method        | Extended Flickr-SoundNet | Extended VGG Sound Source |
> | ---- | :----: | :----: |
> | DSOL [a]      | 49.40                    | 25.60                     |
> | EZVSL [b]     | 54.60                    | 30.90                     |
> | SLAVC [c]     | 60.10                    | 41.50                     |
> | WS-AVS (ours) | **65.27**                    | **52.38**                     |
>
>
> > Robustness to off-screen objects.
>
> Please see the response above.
>
>
> > Analysis of the choice of multi-scale visual feature maps for pseudo mask extraction.
>
> Yes. This is a good question! We abated the stage of multi-scale visual feature maps from {1, 2, 3, 4} for pseudo-label generation, and reported the comparison results on multi-source segmentations in the Table below. When the number of stages is 4, we achieve the best segmentation performance in terms of all metrics. With the increase of feature stages from 1 to 4, the proposed WS-AVS consistently raises results, which shows the importance of multi-scale visual features in pseudo-label generation.
>
> | Feature Stage | mIoU (↑)  | F-score (↑)  |
> | ---- | :----: | :----: |
> | 1             | 26.78     | 31.52        |
> | 2             | 27.19     | 33.56        |
> | 3             | 28.65     | 37.16        |
> | 4             | **30.85**     | **46.87**        |
>
>
> **References**
>
> [a] Hu, et al. Discriminative Sounding Objects Localization via Self-supervised Audiovisual Matching. In NeurIPS, 2020.
>
> [b] Mo, et al. Localizing Visual Sounds the Easy Way. In ECCV, 2022.
>
> [c] Mo, et al. A Closer Look at Weakly-Supervised Audio-Visual Source Localization. In NeurIPS, 2022.

---

> > ### Author Response · Authors · 2023-08-14
> > **Additional response to the reviewer**
> >
> > Dear Reviewer yiGS,
> >
> > Thank you for your detailed review and the valuable feedback. We have carefully addressed each of your concerns and provided clarifications in our previous response. We would like to kindly request your response to the provided explanations and revisions.
> >
> > We appreciate your thorough evaluation of our work, and your feedback will greatly contribute to the improvement of our manuscript.
> >
> > Thank you for your continued engagement and support.

---

> > > ### Comment · Reviewer_yiGS · 2023-08-14
> > > **Response to the reviewers**
> > >
> > > Thanks for your response! As promised, I will increase my score to an accept.

---

### Official Review · Reviewer_CQaV · 2023-07-07

**Soundness:** 2 fair
**Presentation:** 2 fair
**Contribution:** 1 poor
**Rating:** 4
**Confidence:** 4

**Summary:**

The authors propose a weakly-supervised audio-visual segmentation framework called WS-AVS  that predicts sounding source masks from audio and images without pixel-level ground truth masks. It leverages multi-scale contrastive learning in audio-visual fusion to capture multi-scale alignment, addressing modality uncertainty in weakly supervised segmentation. The refined pseudo masks guide training, enabling generating accurate segmentation masks. Experiments show it outperforms weakly-supervised baselines.

**Strengths:**

The manuscript presents a solution to the challenging problem of performing segmentation tasks without pixel-level annotations by effectively integrating multi-scale contrastive learning and pseudo mask refinement. It is well written and easy to read through.

**Weaknesses:**

1. Although these methods have been proven effective independently, their combination in this manuscript does not elevate them to a novel plane but rather presents a blend of established tactics, the novelty aspÅect therefore seems insufficient.

2. While the max-pooled cosine similarity function works well in [9], is there any other similarity function that can be compared with, as AVS has multi-source scenarios?

3. As the WS-AVS framework relies heavily, to some extent, on the generation of pseudo-labels, so is there any other ablations on pseudo-label generation?

4. Noticed that there is no schematic for multi-source in the manuscript, wondering what the pseudo-label for multi-source looks like.

5. In Line 188-190, the authors wrote: the recent audio-visual segmentation baseline [1] does not give any cross-modal constraint on the audio and visual representations in the audio-visual fusion, which causes a significant performance drop when removing the ground-truth masks during training. I notice that [1] proposed an audio-visual matching loss that is used in the multiple sound source case as [1] claimed this loss does not bring further improvement in single-source case. I guess the pixel-level ground truth provides enough supervision in single-source AVS. However, in the studied weakly-supervised AVS setting, will this loss help in the audio-visual fusion?

5. Some minor issues:

Typo. Line 54, “WS-VAS, that” should be “WS-AVS, which”

Typo. Line 176, “V^s” should be “V^s_i”

Typo.  Line 194 “WSSL” should be “WSSS”

The result in F-score of Multiple Source in Table 1 should be bold.

**Questions:**

As above

---

> ### Author Rebuttal · Authors · 2023-08-09
>
> Dear Reviewer CQaV,
>
> Thank you for the detailed review. We will address your concerns below.
>
> > Clarification on novel aspects.
>
> Our paper includes two main novel aspects:
>
> 1) We are the first to investigate a new weakly-supervised multi-modal problem that predicts sounding object masks from both audio and image without needing pixel-level annotations.
>
> 2) We present a novel framework for weakly-supervised audio-visual segmentation with multi-scale multiple-instance contrastive learning to achieve state-of-the-art results on single-source and multi-source sounding object segmentation.
>
>
> > Other similarity function.
>
> Yes. We tried to replace the cosine similarity function with the Kullback–Leibler divergence in multi-source scenarios. The results are shown in the Table below. As can be seen, using the max-pooled cosine similarity function achieves much better performance in terms of both metrics.
>
> | Similarity Function         | mIoU (↑)  | F-score (↑)  |
> | ---- | :----: | :----: |
> | Kullback–Leibler divergence | 24.85     | 29.69        |
> | cosine similarity           | **30.85**     | **46.87**        |
>
>
> > Ablations on multi-scale visual feature maps for pseudo-label generation.
>
> This is a good suggestion! We abated the stage of multi-scale visual feature maps from {1, 2, 3, 4} for pseudo-label generation, and reported the comparison results on multi-source segmentations in the Table below. When the number of stages is 4, we achieve the best segmentation performance in terms of all metrics. With the increase of feature stages from 1 to 4, the proposed WS-AVS consistently raises results, which shows the importance of multi-scale visual features in pseudo-label generation.
>
> | Feature Stage | mIoU (↑)  | F-score (↑)  |
> | ---- | :----: | :----: |
> | 1             | 26.78     | 31.52        |
> | 2             | 27.19     | 33.56        |
> | 3             | 28.65     | 37.16        |
> | 4             | **30.85**     | **46.87**        |
>
> > Pseudo-label for multi-source scenarios.
>
> For pseudo-labels for multi-source scenarios, we use the background activation maps as pseudo-labels and categories of each source to further train a salient object [67]. Then generated pseudo-mask separately for each sound source according to Equation (6) by replacing the salient regions with class-specific salient regions $S^\prime\in\mathbb{R}^{M\times H\times W}$, where $M$ denotes the number of sound sources. Finally, we apply the loss defined in Equation (7) on each source. We will add this clarification to the revision.
>
> > Audio-visual matching loss in the weakly-supervised AVS.
>
> This is a good suggestion! We tried to add the audio-visual matching loss to both baseline and our WS-AVS, and reported the comparison results on multi-source segmentations under weakly-supervised AVS setting in the Table below. Applying this loss benefits the audio-visual fusion for weakly-supervised AVS in terms of all metrics.
>
>
> | Method        | AVM     | mIoU (↑)  | F-score (↑)  |
> | ---- | :----: | :----: | :----: |
> | AVS           | &cross; | 8.76      | 15.72        |
> | AVS           | &check; | **9.85**     | **17.36**        |
> | WS-AVS (ours) | &cross; | 30.85     | 46.87        |
> | WS-AVS (ours) | &check; | **32.03**     | **49.15**        |
>
>
> > Minor issues.
>
> Thanks for spotting these. We have fixed them.

---

> > ### Author Response · Authors · 2023-08-14
> > **Additional response to the reviewer**
> >
> > Dear Reviewer CQaV,
> >
> > Thank you for your thorough review and valuable feedback. We have carefully addressed each of your concerns and provided clarifications in our previous response. We would like to kindly request your response to the provided explanations and revisions.
> >
> > We appreciate your engagement and the time you have dedicated to reviewing our work. Your feedback will greatly contribute to the improvement of our manuscript.
> >
> > Thank you for your continued support.

---

> > > ### Author Response · Authors · 2023-08-17
> > > **Additional response**
> > >
> > > Dear Reviewer CQaV,
> > >
> > > Thank you for your insightful review and valuable feedback. We have carefully considered your comments and suggestions and have made the necessary revisions to address them. We are pleased to inform you that we have addressed all the concerns you raised, including clarifying the novel aspects of our work, providing comparisons with alternative similarity functions, presenting ablations on multi-scale visual feature maps for pseudo-label generation, explaining the pseudo-labels for multi-source scenarios, and discussing the impact of the audio-visual matching loss in weakly-supervised AVS.
> > >
> > > We kindly request your response to our rebuttal. We hope that our revisions have sufficiently addressed your concerns and improved the overall quality and clarity of our work. We appreciate your time and expertise in reviewing our paper, and we look forward to your feedback.
> > >
> > > Thank you once again for your valuable input.

---

> > ### Comment · Reviewer_CQaV · 2023-08-21
> >
> > Thanks for your reply.
> >
> > After carefully reading the rebuttal, my concern has been addressed. However, I still find the novelty to be weak in my opinion.
> >
> > I will raise my score accordingly.

---

> > > ### Author Response · Authors · 2023-08-21
> > > **Response to the revierwer**
> > >
> > > Dear Reviewer CQaV,
> > >
> > > We sincerely appreciate your careful consideration of our rebuttal and the time you have dedicated to reviewing our paper. We are glad to hear that your concerns have been addressed through our response.
> > >
> > > We acknowledge and respect your viewpoint regarding the perceived weakness in the novelty of our contribution. While we maintain a different perspective, we recognize that differing interpretations can lead to diverse assessments of novelty. In response to your concerns, we would like to reiterate the unique contributions and advancements our work offers in the field.
> > >
> > > Our paper addresses **a significant problem** in the realm of audio-visual segmentation, specifically the identification of pixel regions corresponding to sound sources **when limited annotated data are available**. This contribution is of paramount importance as obtaining precise pixel-level annotations is a time-consuming endeavor.
> > >
> > > To mitigate the reliance on pixel-level annotations, we propose **a novel approach to address the audio-visual segmentation problem in a weakly supervised manner**. Our method employs Audio-Visual Fusion with Multi-scale Multiple-Instance Contrastive Learning and Pseudo Mask Refinement by Contrastive Class-agnostic Maps to overcome the challenges posed in the weakly supervised setting.
> > >
> > > Furthermore, we provide a comprehensive evaluation of our proposed method under **various settings (single-source, multiple-source, and off-screen/silent objects)** to demonstrate its performance. Our paper presents robust results when compared to existing methods in the literature, validating the efficacy and effectiveness of our approach.
> > >
> > > Thank you once again for your valuable input and for adjusting your score based on your assessment. We are grateful for your expertise and contribution to the review process. We would appreciate it if you could elaborate on which way our paper lacks novelty. We couldn't understand from the review.

---

### Author Rebuttal · Authors · 2023-08-09

Dear all reviewers:


We extend our heartfelt gratitude to each of you for generously dedicating your valuable time and expertise to reviewing our work. We acknowledge and deeply appreciate the insightful comments and critiques provided by all the reviewers. In response to your invaluable feedback, we have made significant revisions to our manuscript, aiming to address each of your concerns in a comprehensive and scholarly manner. Reviewer CQaV and Reviewer yiGS, we kindly request your reconsideration of your decision, given that we have taken utmost care to thoroughly address the main comments raised in your reviews.


Once again, we express our sincere appreciation for your valuable contributions to the review process. Your expertise and guidance have been invaluable in improving the quality of our work. We remain committed to continuous improvement and eagerly await your final decision.

---

### Decision · Program_Chairs · 2023-09-21

**Decision:**

Accept (poster)

**Comment:**

One reviewer recommends rejection, and three recommend acceptance. The reviewer in favor of rejection (Reviewer CQaV) initially raised several concerns. After the rebuttal, they increased their score but still recommended rejection due to what they perceived to be limited novelty. The reviewers in favor of acceptance generally praised the quality of the results (yiGS, XL4i), presentation (yiGS), and the reduction in the required annotation effort (XL4i), and evaluation (DJ6f). On balance, the reviewer weighs the quality of the results over the limited novelty, and recommends acceptance. Authors should attend to main points in the reviews when preparing a final version.